# Short-Term Adverse Effects Immediately after the Start of COVID-19 Booster Vaccination in Vietnam

**DOI:** 10.3390/vaccines10081325

**Published:** 2022-08-16

**Authors:** Duy Cuong Nguyen, Thi Loi Dao, Thi Minh Dieu Truong, Thu Huong Nguyen, Thu Nga Phan, Ha My Nguyen, Thi Dung Pham, Xuan Bai Nguyen, Thanh Binh Nguyen, Van Thuan Hoang

**Affiliations:** Thai Binh University of Medicine and Pharmacy, Thai Binh 410000, Vietnam

**Keywords:** vaccine, COVID-19, booster, side effect

## Abstract

Background: Risk communication is necessary to improve the booster vaccination rate, but Vietnam does not have a system to collect and disclose such information. Therefore, the purpose of this study was to clarify adverse reactions and their frequency in the early period after booster vaccination, and to obtain primary data for improving the booster vaccination rate. Methods: A cross-sectional survey was conducted among adults aged ≥18 years. Clinical data were collected 14 days after booster vaccination by using a standard questionnaire. Results: A total of 1322 participants were included with median age = 23 and sex ratio (Male/Female) = 0.53. AstraZeneca was the most commonly used vaccine for the first and second doses, while Pfizer was the most commonly used vaccine for booster shots. Injection site pain, fatigue, and myalgia were the most common side effect reported (71.9%, 28.1%, and 21.8%, respectively). Compared to previous COVID-19 vaccine injections, 81.9% of participants reported that their symptoms were similar or milder after receiving the booster dose. They were more likely to present injection site pain (OR = 1.43, *p* < 0.0001) and lymphadenopathy (OR = 4.76, *p* < 0.0001) after receiving the booster shot. Fever (OR = 0.33, *p* < 0.0001) and fatigue (OR = 0.77, *p* = 0.002) were less often reported after booster shots compared to the first and second injections. The severity of symptoms occurring after booster dose versus first and second doses increased significantly with each additional year of age and among participants receiving the Pfizer and Moderna vaccines. Conclusion: Adverse reactions to booster vaccination are minor and their incidence is the same as for the first or the second vaccination. Multicenter studies with larger sample sizes on the side effects and safety of COVID-19 vaccine booster shots need to be conducted to make the population less worried, in order to increase the vaccination rate, to protect individuals’ and communities’ health.

## 1. Introduction

Vaccines against COVID-19 are considered the most effective specific preventive measure. On 31 December 2020, Pfizer was the first vaccine to be approved by the WHO for emergency use on a global scale [1]. Since then, 10 vaccines against COVID-19 have been approved by the WHO for use [2]. A primary vaccination course consists of two doses to achieve protective antibody titers. However, recent reports increasingly show that the effectiveness of the COVID-19 vaccine is reduced within 6 months after the deployment of the primer dose [3,4,5]. The COVID-19 vaccine booster dose could further restore and enhance protection that may have decreased over time following the primary series immunization. Individuals with an updated COVID-19 vaccine status are best protected from severe COVID-19 disease. Therefore, booster shots are essential in the context of the ongoing pandemic due to new variants of concerns. As of 4 August 2022, 12.4 billion doses of the vaccine have been administered worldwide, and 6.3 million doses are given each day. Over 67.2% of the world’s population has received at least one dose of the COVID-19 vaccine [6]. It can be said that this has been the largest vaccination campaign on a global scale ever.

Although non-pharmaceutical measures and a large number of vaccines against COVID-19 have been used, the COVID-19 pandemic has yet remained uncontrolled, partly due to the ability of the virus to mutate. Furthermore, the appearance of variants of concerns such as Alpha, Beta, Gamma, Delta, and most recently Omicron, with their ability to rapidly spread and evade the immune system, makes the control of the disease even more challenging [7]. Viral mutations play a crucial role in the loss of neutralizing activity of antibodies, requiring additional or booster doses to maintain antibody titers [8].

Like any other vaccine, the COVID-19 vaccine also has side effects. Most of these side effects are mild, mainly pain at the injection site, fever, muscle aches, headache, and fatigue, and they usually disappear after a few days without treatment. However, there are also some other undesirable effects, which are rarer, but manifest in severe diseases, such as anaphylaxis, coagulation, myocarditis, thyroiditis, and even death [9]. This is the reason why people are concerned about not getting vaccinated against COVID-19.

In Vietnam, the vaccination campaign against COVID-19 started on March 8, 2021 [10]. Four-fifths (83.6%) of the total population were completely vaccinated with the primary (first and second) doses. However, the proportion of people receiving booster shots is still very limited even though the injection has been administered since December 2021. At the time of writing, only 17% of the population has received it [10]. The main reasons for refusing this booster dose of the COVID-19 vaccine might include adverse experiences of previous injections, the conception that additional vaccinations are unnecessary because the COVID-19 pandemic is under control, or uncertainty of the safety of the vaccine [11]. Indeed, in a survey conducted by Rzymski et al., half of the reasons for refusing booster doses were side effects experienced after the previous COVID-19 vaccine. A quarter of participants who refused said, the safety of vaccines was uncertain [11]. An online survey in Poland showed that the most frequent reason why the participants refused to receive the booster dose was the side effects it might cause to themselves or their loved ones [12]. A proportion of 24.6% of respondents reported that they could not tolerate the booster injection because the adverse reaction of the previous ones was severe [13]. In an Italian study conducted in November and December 2021, 26.1% of participants worried about serious side effects occurring after a booster dose of the COVID-19 vaccine [14].

Risk communication is necessary to improve the booster vaccination rate, but Vietnam does not have a system to collect and disclose such information. Therefore, we conducted this study to clarify adverse reactions and their frequency in the early period after booster vaccination and to obtain primary data for improving the booster vaccination rate.

## 2. Materials and Methods

### 2.1. Study Population and Questionnaire Design

In Vietnam, the booster dose of the COVID-19 vaccine has been introduced since December 10, 2021. Individuals aged 18 years old or older who have received the primary or additional dose could receive this injection. Persons with chronic diseases or who need long-term care at medical facilities, aged 50 years or older, healthcare workers who have direct contact with COVID-19 patients, and medical staff are given priority for the booster dose [15]. The Vietnamese government also regulates the type of vaccine for booster shots as follows: If the primary or additional shots were the same vaccine, the booster will be either the vaccine of the same type or the mRNA vaccine. If the primary shots were admitted with different vaccines, the booster dose will use the mRNA vaccine. If the primary or additional dose is a vaccine of Sinopharm, a booster shot of the same type or an mRNA vaccine, or a viral vector vaccine (e.g., AstraZeneca vaccine) can be given. The booster dose is at least 6 months after the last dose of the primary or additional dose [15].

This single-center study was conducted on all individuals who visited Thai Binh University of Medicine and Pharmacy to receive a booster dose of vaccine from 1 January to 28 February, right after the start of the program. This cross-sectional survey was conducted at Thai Binh University of Medicine and Pharmacy, Thai Binh, Vietnam from 1 January 2022 to 28 February 2022. Adults in the age group of 18 years or older were eligible for participation. Data were collected by using a standard questionnaire which includes 4 sections: characteristics of participants (age, gender, chronic diseases) and clinical side effects of COVID-19 vaccines after the first, second, and booster doses.

During the consultation for booster vaccination, participants were offered the chance to participate in our study. They were clearly informed about the aims of the study, the number of questions, and how the collected data will be stored. They were also informed that this study did not use personal identities and the collected data were used for research purposes only. They had the right to refuse or voluntarily participate in the survey. Informed consent was obtained from respondents before completing the questionnaire.

Persons who consented to participate were asked to complete their clinical signs for 14 days after receiving the COVID-19 vaccine. Data on the side effects of each dose were retrospectively collected after receiving the booster injection.

### 2.2. Data Analysis

R version 4.2.1 (R Foundation for Statistical Computing, Vienna, Austria) (https://cran.r-project.org/, accessed on 1 July 2022) was used for statistical analyses. All the qualitative variables were presented by number and percentage. The quantitative variables were presented as median, interquartile, and range. The Chi-square test for R by C table was used to evaluate the difference in proportions of COVID-19 vaccine, side effects, and severity of symptoms between three groups: first dose, second dose, and the booster dose. The main outcome was COVID-19 vaccine side effects between the booster versus first dose and second dose. The second outcome was clinical symptoms occurring after each vaccination. A comparison between the booster and primary dose for the same symptom and the association between age, gender, medical history, type of COVID-19 vaccine, and clinical symptoms were analyzed using bivariate analysis. Variables with *p*-value < 0.2 in the bivariate analysis were imported into the multivariate analysis using logistic regression [16]. The results were presented as OR (odds ratio) and 95% CI (95% confidence interval). A *p*-value < 0.05 was considered significant.

## 3. Results

### 3.1. Characteristics of the Included Population

During the study period, 1467 individuals were vaccinated with a booster dose and invited to participate in the study. A total of 1322 persons (90.1%) agreed to participate and were eligible for selection. The median age was 23 years (range = 18 to 84 years). In total, 65.4% of people (864/1322) were female, hence the sex ratio (Male/Female) was 0.53. High blood pressure was the most frequent chronic condition of the participants (39/1322, 3.0%), followed by diabetes (16/1322, 1.2%), chronic heart diseases (10/1322, 0.8%), and chronic respiratory diseases (8/1322, 0.6%) (Table 1).

### 3.2. COVID-19 Vaccines and Side Effects Reported by Participants

mRNA vaccines were used to vaccine 360 (27.2%), 369 (27.9%), and 866 (65.5%) persons for the first, second, and booster doses, respectively.

After the first dose of the COVID-19 vaccine, injection site pain was the most frequent (74.4%) side effect reported by participants, followed by fever (49.8%), fatigue (48.6%), and myalgia (39.9%). In total, 63.8% of participants reported that they had injection site pain after the second dose of COVID-19 vaccination, while 31.6% and 27.0% presented fatigue and fever, respectively. After the COVID-19 booster dose, injection site pain was still the most common side effect (71.9%), followed by fatigue (28.1%) and myalgia (21.8%). None of the participants required medical care for their symptoms after receiving the booster dose.

Compared to the previous COVID-19 vaccine injection, 20.7% and 18.1% of participants reported that their symptoms were more severe after the second and booster doses, respectively. However, this difference was not statistically significant, with a *p*-value = 0.09 (Table 2).

### 3.3. Factors Associated with the Presence of Side Effects after Receiving Each Dose of COVID-19 Vaccine

The Appendix A showed the factor associated with the presence of side effects after receiving each dose of the COVID-19 vaccine. The incidence of adverse reactions after the first dose decreased significantly with each year of age (OR = 0.97, *p*-value < 0.0001). Male gender was less likely to have clinical symptoms after receiving all the vaccines (OR = 0.46, *p*-value < 0.0001, OR = 0.47, *p*-value < 0.0001, and OR = 0.57, *p*-value < 0.0001, respectively). Chronic medical conditions did not increase the likelihood of adverse events after vaccination against COVID-19. Finally, compared with persons vaccinated with the AstraZeneca vaccine, those vaccinated with the Sinopharm vaccine were less likely to be exposed to side effects after the first and second doses (OR = 0.09, *p*-value < 0.0001, and OR = 0.42, *p*-value < 0.0001). However, those vaccinated with the Pfizer and Moderna vaccines were more likely to present side effects after the second dose (OR = 3.54, *p*-value < 0.0001, and OR = 9.29, *p*-value < 0.0001, respectively). Participants vaccinated with the Pfizer vaccine were three times more likely to present adverse reactions after their booster dose (OR = 3.21, *p*-value < 0.0001).

### 3.4. Comparison of COVID-19 Vaccine Side Effects between Booster versus First Dose and Second Dose

Table 3 shows the comparison of COVID-19 vaccine side effects between the booster versus the first dose and second dose. After the booster dose injection, the participants were more likely to present injection site pain and lymphadenopathy with adjusted OR (aOR) = 1.43, 95%CI = [1.22–1.67], and aOR = 4.76, 95%CI = [3.39–6.67]. However, fever (aOR = 0.33, 95%CI = [0.27–0.39]) and fatigue (aOR = 0.77, 95%CI = [0.65–0.91]) were less often reported after booster shots compared to the first and second injections.

Table 4 shows factors associated with the severity of side effects occurring after the booster dose versus the first and second doses. The severity of symptoms increased significantly with each additional year of age with OR = 1.02, *p*-value < 0.0001. In addition, participants receiving the Pfizer and Moderna vaccines were four and nine times more likely to experience serious side effects than those receiving the AstraZeneca vaccine (OR = 3.78, *p*-value < 0.0001, and OR = 9.46, *p*-value < 0.0001, respectively).

## 4. Discussion

Studies on the side effects of vaccines against COVID-19 are currently necessary. They provide knowledge and understanding to devise strategies for public health care in the post-vaccination period, as well as the timely detection and resolution of severe complications after the vaccination campaign. Nearly 2 years after the WHO approved the first COVID-19 vaccine, with more than 67% of the population worldwide receiving at least one dose, the short-term adverse effects of the primary dose of COVID-19 vaccine are well known [9,17]. However, studies on the adverse effects of booster doses remain limited, or conducted in special populations, including the elderly and nursing home residents [18,19,20,21,22,23,24].

Additionally, studies on the side effects of the COVID-19 vaccine booster injection help to remove worries as well as reduce the hesitation of the population towards this injection. Indeed, in a recent study, only 71% of people have been willing to get a booster shot. The main reasons why they refused vaccinations are the side effects and the uncertain safety of vaccination [11]. Currently, in Vietnam, there is no system to survey events occurring after COVID-19 vaccination in general, and booster shots in particular. Therefore, it is difficult to collect and monitor adverse symptoms after vaccination.

Our study shows that the side effects of the COVID-19 vaccine booster shot are similar to those of the previous injections. In our survey, only six and two persons required medical care after the first and second doses of COVID-19 vaccine, respectively. It should be noted that no patient had to be hospitalized because of their clinical symptoms after the booster vaccination. Approximately 82% of participants reported that their symptoms after the booster dose did not differ or were milder compared to those of the previous vaccine doses. When analyzing factors related to the presence of adverse reactions after each dose of vaccine, our results showed that the age of participants and male gender were associated with a decrease in side effect incidences after injections. In addition, medical history of chronic disease did not increase the likelihood of adverse events after vaccination against COVID-19. However, the gravity of their side effects after booster shots tended to be more severe, compared with primary vaccinations. Interestingly, older age, male gender, and chronic disease are major risk factors for severe COVID-19 disease [25]. Therefore, these subjects need to be protected by the COVID-19 vaccine to reduce the mortality rate when infected. They also need more care in the post-vaccination period.

Our results are comparable to those of previous studies. In a clinical trial study, the adverse effects of the COVID-19 vaccine booster dose were similar to those after the second dose [20]. From August 12 to 19 September 2021, over 12,500 registrants reported receipt of a booster dose of the COVID-19 vaccine [20]. The findings from one electronic survey of Hause et al. indicated that most of the adverse reactions after receiving this vaccination were mild or moderate [20]. Indeed, 79.4% and 74.1% of participants reported local or systemic reactions, respectively. The proportion of these symptoms was similar to those after the second dose (77.6% and 76.5%, respectively) [21]. In another online survey including 721,562 adult participants receiving the mRNA booster shot, their local and systemic reactions were less frequently reported during the week following this injection than when they received the second dose [22].

In a US cohort surveillance study using electronic health records that included 47,999 participants, a booster dose vaccination against SARS-CoV-2 infection was significantly associated with a low risk of severe adverse events. In addition, the proportion of severe side effects remained comparable to those of the primary regime [24]. Among 27,046 participants aged ≥ 60 years old in Israel who received a booster dose of the Pfizer vaccine, 30% of participants had at least one adverse effect. Injection site pain, fatigue, and malaise were the most frequently reported (23.5%, 9.7%, and 7.2%, respectively) [18]. Only 11.1% of participants presented a worse symptom compared to the second vaccine dose and 1.2% required medical care [18]. In the community-based ZOE COVID study that included 317,011 participants receiving the booster dose, the adverse effects recorded were similar to those after the previous dose [23]. Another survey of 11,200 nursing home residents receiving booster doses reported that no adverse event was observed during the 14-day period after vaccination [19]. In our work, after the booster dose injection, the participants were more likely to present injection site pain and lymphadenopathy, but fever and fatigue are less often reported after booster shots compared to the previous injections.

Our study has some limitations. We did not investigate the history of COVID-19 infection to evaluate its influence on adverse effects after vaccination. This is a single-center study, conducted using a modest sample size, and the studied population was young. Our results might not be representative of the general population. Clinical symptoms related to adverse events of the COVID-19 vaccine were also collected based on the subjective responses of the participants and depended on their recall. Hence, recall bias should be considered. For the first dose of the vaccine, the occurrence of adverse reactions decreased significantly with each additional year of age. It is unclear whether the adverse reactions ceased with increasing age or whether the subjects forgot about the adverse reactions that occurred after the first dose. It is also possible that as age increases, subjects may forget past adverse reactions and remember the most recent adverse reactions. Thus, we cannot significantly rule out the possibility of a recall bias as age increases. In the future, we would like to consider a system to record adverse reactions for each vaccination. We also only examined short-term symptoms without mentioning the duration of side effects. Therefore, this study may not detect severe, rare, and late adverse events following the COVID-19 booster vaccination. However, previous studies showed that COVID-19 vaccine booster shots have fewer serious side effects than previous injections [24,26]. Specifically, in a study conducted by Kuehn et al., myocarditis was uncommon after a booster dose of the COVID-19 vaccine [26]. The confirmed myocarditis incidence after a booster dose was 11.4 per 1 million administered doses among male adolescents. By comparison, this incidence after the second dose was reported as 105.9 per 1 million doses and 70.7 per 1 million among adolescents of 16- to 17-year-olds and 12- to 15-year-olds, respectively [26].

## 5. Conclusions

Adverse reactions to the booster vaccination are minor and their incidence is the same as for the first or the second vaccination. The elderly, persons with chronic diseases, or those vaccinated with mRNA vaccines need to be cared for after booster vaccination to limit the effects of side effects on their health. The world is facing a new wave of the COVID-19 pandemic due to the Omicron BA.4 and BA.5 variants [27]. Vaccines are by far the most effective measure to reduce the mortality rate from COVID-19. Multicenter studies on the side effects and safety of COVID-19 vaccine booster shots with larger sample sizes need to be conducted to make the population less worried, in order to increase the vaccination rate for this injection, to protect individuals’ and communities’ health. A common online monitoring and reporting system in many languages around the world should also be implemented to share information about the safety of the vaccine.

## Figures and Tables

**Table 1 vaccines-10-01325-t001:** Characteristics of the included population.

Characteristics	N = 1322	%
Age (year)		
Median	23	
Interquartile	22–33	
Range	18–84	
Gender		
Female	864	65.4
Male	458	34.6
At least one chronic disease	91	6.9
Chronic respiratory diseases	8	0.6
Chronic heart diseases	10	0.8
High blood pressure	39	3.0
Diabetes	16	1.2
Others	40	3.0

**Table 2 vaccines-10-01325-t002:** COVID-19 vaccines and side effects reported by studied population.

	First Dose N = 1322	Second Dose N = 1322	Booster Dose N = 1322	*p*-Value
COVID-19 vaccine				
mRNA	360 (27.2)	369 (27.9)	866 (65.5)	<0.0001
Others	962 (72.8)	953 (72.1)	456 (34.5)	
Detailed information about type of COVID-19 vaccines				
Pfizer	83 (6.3)	98 (7.4)	836 (63.2)	<0.0001
Moderna	277 (21.0)	275 (20.8)	30 (2.3)	
AstraZeneca	898 (67.9)	885 (66.9)	455 (34.5)	
Sinopharm	63 (4.8)	63 (4.8)	1 (0.0)	
Sputnik V	1 (0.0)	1 (0.1)	0 (0)	
Side effects				
At least one side effect	1141 (86.3)	965 (73.0)	1045 (79.1)	<0.0001
Injection site pain	984 (74.4)	843 (63.8)	951 (71.9)	<0.0001
Fever	658 (49.8)	357 (27.0)	236 (17.9)	<0.0001
Lymphadenopathy	37 (2.8)	28 (2.1)	104 (7.9)	<0.0001
Fatigue	642 (48.6)	418 (31.6)	371 (28.1)	<0.0001
Headache	390 (29.5)	238 (18.0)	260 (19.7)	<0.0001
Vertigo	134 (10.1)	76 (5.8)	79 (6.0)	<0.0001
Nausea, vomiting	51 (3.9)	26 (2.0)	29 (2.2)	0.004
Myalgia	527 (39.9)	280 (21.2)	288 (21.8)	<0.0001
Joint pain	132 (10.0)	76 (5.8)	78 (5.9)	<0.0001
Others	50 (3.8)	38 (2.9)	45 (3.4)	0.43
Requiring medical care	6 (0.5)	2 (0.2)	0 (0)	0.03
Severity of symptoms compared to the previous injection				
More severe	-	274 (20.7)	239 (18.1)	0.09
Milder or equivalent	-	1048 (79.3)	1083 (81.9)	

**Table 3 vaccines-10-01325-t003:** Comparison of COVID-19 vaccine side effects between the booster versus the first dose and second dose: bivariate and multivariate analyses.

	Univariate Analysis			Multivariate Analysis *		
	OR	95%CI	*p*-Value	Adjusted OR	95%CI	*p*-Value
None	1.04	0.88–1.22	0.66			
Fever	0.35	0.30–0.41	<0.0001	0.33	0.27–0.39	<0.0001
Fatigue	0.58	0.51–0.67	<0.0001	0.77	0.65–0.91	0.002
Myalgia	0.63	0.54–0.74	<0.0001			
Injection site pain	1.14	0.99–1.33	0.07	1.43	1.22–1.67	<0.0001
Lymphadenopathy	3.39	2.47–4.65	<0.0001	4.76	3.39–6.67	<0.0001
Headache	0.79	0.67–0.92	0.004			
Vertigo	0.74	0.56–0.96	0.03			
Nausea, vomiting	0.75	0.49–1.15	0.19			
Joint pain	0.73	0.56–0.96	0.02			
Others	1.02	0.71–1.48	0.90			

*: logistic regression; only variables with *p*-value < 0.2 were introduced in the model.

**Table 4 vaccines-10-01325-t004:** Factors associated with the severity of side effects after receiving booster dose versus first and second doses of COVID-19 vaccine.

	Bivariate Analysis			Multivariate Analysis *		
	OR	95%CI	*p*-Value	Adjusted OR	95%CI	*p*-Value
Age (year)	1.03	1.02–1.04	<0.0001	1.02	1.01–1.04	<0.0001
Gender						
Female	reference			reference		
Male	1.03	0.77–1.38	0.86			
Chronic disease						
No	reference			reference		
Yes	2.40	1.52–3.82	<0.0001	1.30	0.75–2.26	0.35
COVID-19 vaccine (booster dose)						
AstraZeneca	reference			reference		
Sinopharm	NA	NA	NA	NA	NA	NA
Pfizer	4.14	2.78–6.17	<0.0001	3.78	2.53–5.64	<0.0001
Moderna	12.00	5.37–26.82	<0.0001	9.46	4.18–21.42	<0.0001

* logistic regression; only variables with *p*-value < 0.2 were introduced in the model; NA: not applicable because only one person was vaccinated with the Sinopharm vaccine for the booster dose.

## Data Availability

The data presented in this study are available on request from the corresponding author (V.T.H.) upon reasonable request.

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
