# Peer review of "Short-Term Adverse Effects Immediately after the Start of COVID-19 Booster Vaccination in Vietnam"

_vaccines, 2022, doi:10.3390/vaccines10081325_

Round 1

Reviewer 1 Report

The manuscript submitted by Duy Cuong Nguyen and collab. aims to evaluate the side effects of the COVID-19 vaccine booster dose. Although the number of participants is high 1322 subjects, the information provided relates strictly to the incidence of side effects after booster dose compared to effects after dose II or I. No information is available on the types of vaccine used, the presence of additional health conditions, or previous diagnosis of COVID-19. I believe that, in its current form, the manuscript does not bring new and useful information to readers.

Improvements to be considered:

Grouping cohort according to vaccination scheme/type of vaccine

Timing of side effects

Duration of the side effects (days)

Medical care needed to mitigate side effects

Page 3 Data analysis “Unadjusted associations between the multiple symptoms were investigated by univariate analysis”. It is unclear how the association between several symptoms can be analyzed by univariate analysis considering the known definitions: Univariate analysis is the analysis of one variable. Multivariate analysis is the analysis of more than one variable. Maybe it is a comparison between the booster and dose II/I for the same symptom.

Author Response

The manuscript submitted by Duy Cuong Nguyen and collab. aims to evaluate the side effects of the COVID-19 vaccine booster dose. Although the number of participants is high 1322 subjects, the information provided relates strictly to the incidence of side effects after booster dose compared to effects after dose II or I. No information is available on the types of vaccine used, the presence of additional health conditions, or previous diagnosis of COVID-19. I believe that, in its current form, the manuscript does not bring new and useful information to readers.

Answer: Thank you very much for your comments. Information on the type of vaccine used has been presented in table 2. The presence and additional health conditions have been introduced in table 1. However, we have no information about the previous diagnosis of COVID-19. We added it now in the limitations of our study.

Improvements to be considered:

Grouping cohort according to vaccination scheme/type of vaccine

Answer: It was added in table 2

Timing of side effects

Answer: Unfortunately, we have no information on the timing of side effects. We added it to the limitation of study.

Duration of the side effects (days)

Answer: Unfortunately, we have no information on the timing of side effects. We added it to the limitation of study

Medical care needed to mitigate side effects

 Answer: It has been presented in table 2 (Requiring medical care)

Page 3 Data analysis “Unadjusted associations between the multiple symptoms were investigated by univariate analysis”. It is unclear how the association between several symptoms can be analyzed by univariate analysis considering the known definitions: Univariate analysis is the analysis of one variable. Multivariate analysis is the analysis of more than one variable. Maybe it is a comparison between the booster and dose II/I for the same symptom.

Answer: We sincerely apologize for this misunderstanding. This sentence was reworded as following:

“The main outcome was COVID-19 vaccine side effects between booster versus first dose and second dose. The second outcome was clinical symptoms occurring after each vaccination. The comparison between the booster and primary dose for the same symptom as well as the association between age, gender, medical history and type of COVID-19 vaccine and clinical symptoms were analyzed using bivariate analysis. Variables with p-value <0.2 in the bivariate analysis were imported into the multivariate analysis using logistic regression. The results were presented as OR (odds ratio) and 95%CI (95% confidence interval). A p-value < 0.05 was considered as significant.”

Reviewer 2 Report

I salute my colleagues who are planning to end the COVID-19 pandemic. However, I am unable to publish your paper as it stands. Below are my concerns and proposed solutions, which I hope you will find helpful. However, you do not have to accept my views in their entirety. If you have any differences, please do not hesitate to let me know. I would be honored to help you improve your paper.

Abstract

1) Background

It says background, but there is no background. Thus, it does not say why this study is necessary. What is written in the "Conclusion" could also serve as background. As a suggestion, the following should be included.

Risk communication is necessary to improve the booster vaccination rate, but Vietnam does not have a system to collect and disclose such information. Therefore, the purpose of this study was to clarify adverse reactions and their frequency in the early period after booster vaccination and to obtain primary data for improving the booster vaccination rate.

2) Conclusion

 It is better to write what can be said from this original paper. For example, adverse reactions to booster vaccination, if any, are minor and their incidence is the same (low) as for the first or the second vaccination,

etc. However, the analysis may need to be revised, so please revise it with new results.

Introduction

3) I have the impression that many items do not need to be included now that COVID-19 is not in its early stages. For example, the eight lines at the beginning.

4) Few citations of an article describing how experiences or concerns about adverse reactions to vaccines have affected subsequent vaccinations.

5) Overlaps with 3) and 4), but should describe clearly and concisely why the objectives should be met.

Methods

6) Is the time and place for achieving the objective appropriate? Are there any matters not mentioned here that the reader may want to know about? For example, the Thai Binh University of Medicine and Pharmacy is where X % of Vietnamese come here to receive the COVID-19 vaccine. The survey was conducted for two months, from January 1 to February 28, right after the start of the program, to increase the vaccination rate of a booster dose, e.t.

7) Was this survey conducted with the approval of the Ethics Review Committee? Or was it planned and conducted following the Declaration of Helsinki?

8) The method for ‘The difference in proportions’ is unclear as it is not stated in the table. was a 2 x 3 χ-square test performed? Is it Symptom Presence (2) x First dose Second dose, Booster dose (3)?

9) Is there any reason why you chose p<0.2 for univariate analysis?

However, if the following 10) is addressed, this problem will disappear.

10) It is unclear what items are included in the multivariate analysis.

If included, gender, age, and medical history would be appropriate, rather than including multiple symptoms.

11) The odds ratios for the booster dose (third time) were calculated for the first and second times, but the first and second times were almost a year ago. Therefore, a large information bias (recall bias) exists. Therefore, do you think it is meaningful to calculate the probability of adverse drug reactions due to booster administration based on such matters?

As a suggestion, I think it would be better to give odds ratios for each person with symptoms (First, Second, Booster) compared to those without symptoms.

Tables

12) Adjustment coefficients, etc. should be shown outside the column so that the contents can be understood by looking at the table.

Discussion

13) I feel that the background, objectives, methods, and results are not consistent. There are many unnecessary items. I feel that each chapter needs to be rewritten after reconsideration so that it leads to the conclusions of this study.

Also, the limitations are not adequately described.

14) The most important problem is the recall bias mentioned in (11), which remains even with the measures in (11). This needs to be noted.

15) I have the impression that the age group is skewed toward younger people, but how does this compare to the Vietnamese population?

Conclusion

16)The original conclusion of this study is not stated.

Author Response

I salute my colleagues who are planning to end the COVID-19 pandemic. However, I am unable to publish your paper as it stands. Below are my concerns and proposed solutions, which I hope you will find helpful. However, you do not have to accept my views in their entirety. If you have any differences, please do not hesitate to let me know. I would be honored to help you improve your paper.

Answer: Thank you very much for your valuable comments that led us to improve our manuscript. This manuscript was amended in accordance with your recommendations and comments.

Abstract

1) Background

It says background, but there is no background. Thus, it does not say why this study is necessary. What is written in the "Conclusion" could also serve as background. As a suggestion, the following should be included.

Risk communication is necessary to improve the booster vaccination rate, but Vietnam does not have a system to collect and disclose such information. Therefore, the purpose of this study was to clarify adverse reactions and their frequency in the early period after booster vaccination and to obtain primary data for improving the booster vaccination rate.

 Answer: Thank you for your suggestion. It was added to the manuscript.

2) Conclusion

 It is better to write what can be said from this original paper. For example, adverse reactions to booster vaccination, if any, are minor and their incidence is the same (low) as for the first or the second vaccination,

etc. However, the analysis may need to be revised, so please revise it with new results.

 Answer: The analysis was revised according to Reviewers’ suggestions and the conclusion was also reworded.

Introduction

3) I have the impression that many items do not need to be included now that COVID-19 is not in its early stages. For example, the eight lines at the beginning.

Answer: It was removed. 

4) Few citations of an article describing how experiences or concerns about adverse reactions to vaccines have affected subsequent vaccinations.

 Answer: It was added in the introduction

5) Overlaps with 3) and 4), but should describe clearly and concisely why the objectives should be met.

 Answer: The introduction was now revised to clarify the objectives

Methods

6) Is the time and place for achieving the objective appropriate? Are there any matters not mentioned here that the reader may want to know about? For example, the Thai Binh University of Medicine and Pharmacy is where X % of Vietnamese come here to receive the COVID-19 vaccine. The survey was conducted for two months, from January 1 to February 28, right after the start of the program, to increase the vaccination rate of a booster dose, e.t.

Answer: This single-center study was conducted on all individuals who visited Thai Binh University of Medicine and Pharmacy to receive a booster dose of vaccine from January 1 to February 28, right after the start of the program. We added it now in the Methods section.

However, unfortunately, we do not know the number of people vaccinated during this time period at the study site that corresponds to how many percent of Vietnamese people received booster shots. We added it now in the limitation section

7) Was this survey conducted with the approval of the Ethics Review Committee? Or was it planned and conducted following the Declaration of Helsinki?

 Answer: It has been added in Institutional Review Board Statement at the end of manuscript

8) The method for ‘The difference in proportions’ is unclear as it is not stated in the table. was a 2 x 3 χ-square test performed? Is it Symptom Presence (2) x First dose Second dose, Booster dose (3)?

 Answer: R by C table (2 by 3 table in our case) is used to test for an association between variables with more than 2 values. The result is a chi square testing whether the results differ from those expected from the marginal sums alone. The sentence was corrected as below:

“The Chi square test for R by C table was used to evaluate the difference in proportion of COVID-19 vaccine, side effects and severity of symptoms between three groups: first dose, second dose and booster dose.”

9) Is there any reason why you chose p<0.2 for univariate analysis?

However, if the following 10) is addressed, this problem will disappear.

Answer: In our analysis, we have taken the cut-off of 0.2 which was usually used in medical statistic. We added also a reference for this choice.

10) It is unclear what items are included in the multivariate analysis.

If included, gender, age, and medical history would be appropriate, rather than including multiple symptoms.

Answer: because the main outcome of our study was COVID-19 vaccine side effects between booster versus first dose and second dose. The age, sex and chronic diseases of the study participants were similar when compared between these injections. Therefore, we did not include them in the analytical model. However, at your suggestion (11), we analyzed odds ratios for each person with symptoms (First, Second, Booster) compared to those without symptoms. In this supplementary analysis, age, gender and medical history were included.

11) The odds ratios for the booster dose (third time) were calculated for the first and second times, but the first and second times were almost a year ago. Therefore, a large information bias (recall bias) exists. Therefore, do you think it is meaningful to calculate the probability of adverse drug reactions due to booster administration based on such matters?

As a suggestion, I think it would be better to give odds ratios for each person with symptoms (First, Second, Booster) compared to those without symptoms.

 Answer: Thank you very much for your suggestion. We added now this analysis in the manuscript (please see supplementary table 1, 2 and 3)

Tables

12) Adjustment coefficients, etc. should be shown outside the column so that the contents can be understood by looking at the table.

 Answer: It was corrected.

Discussion

13) I feel that the background, objectives, methods, and results are not consistent. There are many unnecessary items. I feel that each chapter needs to be rewritten after reconsideration so that it leads to the conclusions of this study.

Answer: It was revised.

Also, the limitations are not adequately described.

Answer: The limitation was completed.

14) The most important problem is the recall bias mentioned in (11), which remains even with the measures in (11). This needs to be noted.

 Answer: It was noted in limitation

15) I have the impression that the age group is skewed toward younger people, but how does this compare to the Vietnamese population?

 Answer: Yes, studied population was young. Unfortunately, we have no information of vaccinated population in Vietnam to compare. We added it now in the limitation.

Conclusion

16)The original conclusion of this study is not stated.

Answer: The conclusion was revised

Reviewer 3 Report

Nguyen et al performed a retrospective analysis of a cohort of COVID-19 booster vaccine recipients in Vietnam. By comparing the side effects reported by participants between the primary vaccine and the booster, the authors found that booster vaccination is associated with a milder adverse event. This study demonstrates similar findings to prior studies and fortifies public confidence in the safety of booster vaccination.

1.      It is unclear how and when the side effects of 1st and 2nd doses were reported by participants. Was the data collected post each dose or after receiving the booster vaccine? Please clarify this in the methodology.

2.      Please unify the definition of “previous vaccination”, in some places, it refers to 1st while, in other places, it refers to 1st & 2nd doses

3.      Conclusion sections in both abstract and discussion look more like a future direction. It might be better to summarize the findings and propose a brief suggestion accordingly.

4.      It might be out of the scope of this study. I am curious whether the chronic disease impacts the side effect of the COVID19 vaccine? And, how about age and gender?  

5.      Typo and grammar correction. For example, a missing period in the abstract, grammar issues in the third paragraph of the introduction, etc.  

Author Response

Nguyen et al performed a retrospective analysis of a cohort of COVID-19 booster vaccine recipients in Vietnam. By comparing the side effects reported by participants between the primary vaccine and the booster, the authors found that booster vaccination is associated with a milder adverse event. This study demonstrates similar findings to prior studies and fortifies public confidence in the safety of booster vaccination.

  1. It is unclear how and when the side effects of 1st and 2nd doses were reported by participants. Was the data collected post each dose or after receiving the booster vaccine? Please clarify this in the methodology.

Answer: The data on side effects of first and second doses were reported by participants were collected after receiving the booster vaccine. We clarified it now in the Method section

  1. Please unify the definition of “previous vaccination”, in some places, it refers to 1stwhile, in other places, it refers to 1st & 2nd doses

Answer: Thank you for your comment. It was corrected.

  1. Conclusion sections in both abstract and discussion look more like a future direction. It might be better to summarize the findings and propose a brief suggestion accordingly.

Answer: It was reworded.

  1. It might be out of the scope of this study. I am curious whether the chronic disease impacts the side effect of the COVID19 vaccine? And, how about age and gender?  

Answer: Thank you for your comment. We added the associated factor with side effect of COVID-19 vaccine after each dose in which age, gender, chronic disease and type of vaccine were included.

  1. Typo and grammar correction. For example, a missing period in the abstract, grammar issues in the third paragraph of the introduction, etc.  

Answer: it was corrected

Reviewer 4 Report

Estimated Authors,

I've read with interest the present paper on the Short-term effects of COVID-19 booster dose in a study population from Vietnam.

The present study may be of great interest as a large share of the study population received a vaccine primer based on Sinopharm formulate, rather than on other formulates, and particularly mRNA ones. In other words, the present study may share a substantial piece of information on potential consequences of booster doses for population residing in areas where non-mRNA formulates were largely employed (e.g. South America).

Unfortunately, the present paper cannot be accepted for publication for a series of major and minor shortcomings.

To begin with, the statistical analysis can hardly handle the data that were processed. More precisely, the study population seemly considered all the 1322 participants as a single and monolithic group, but (see Table 1) 360 of them were vaccinated with mRNA, and their number increased to 369 for the second dose, and to 866 for the third one. A more appropriate approach would distinctively assess the occurrence of unwanted effects in people having been vaccinated with a mRNA primer (first group), with Sinopharm or Adenovirus-based formulates (group 2 and 3), in order to compare the occurrence of side effects in people having received an homogenous rather than heterogenous ("mosaic") vaccination strategy.

Second, Authors must report how many individuals were initially vaccinated, in order to understand whether the 1322 may acknowledged as representative or not of the original population. 

The minor shortcomings are mainly related to the introduction, as (page 1, since "A primary ..." to "...ever") it mainly deal with the background of late 2021, but an up-ta-date description of the 2022 vaccination campaign must be provided and accordingly discussed.

Author Response

Estimated Authors,

I've read with interest the present paper on the Short-term effects of COVID-19 booster dose in a study population from Vietnam.

The present study may be of great interest as a large share of the study population received a vaccine primer based on Sinopharm formulate, rather than on other formulates, and particularly mRNA ones. In other words, the present study may share a substantial piece of information on potential consequences of booster doses for population residing in areas where non-mRNA formulates were largely employed (e.g. South America).

Answer: Thank you for your comment. Unfortunately, there are only 63 persons were initially vaccinated with Sinopharm vaccine and the sample size was not large enough to conduct an independent analysis. However, as suggestions of Reviewers #2 and #3, we have added an analysis of adverse effects of the COVID-19 vaccine after each injection. The type of vaccine was included in the analysis model.

Unfortunately, the present paper cannot be accepted for publication for a series of major and minor shortcomings.

To begin with, the statistical analysis can hardly handle the data that were processed. More precisely, the study population seemly considered all the 1322 participants as a single and monolithic group, but (see Table 1) 360 of them were vaccinated with mRNA, and their number increased to 369 for the second dose, and to 866 for the third one. A more appropriate approach would distinctively assess the occurrence of unwanted effects in people having been vaccinated with a mRNA primer (first group), with Sinopharm or Adenovirus-based formulates (group 2 and 3), in order to compare the occurrence of side effects in people having received an homogenous rather than heterogenous ("mosaic") vaccination strategy.

Answer: Thank you very much for your comments.

Like other developing countries in the world, in the early stages of the COVID-19 pandemic, Vietnam did not have many vaccines. The Vietnamese government has adopted the policy: " The importance is to be vaccinated as soon as possible with available COVID-19 vaccine once it's your turn and not wait". Therefore, as mentioned in the Methods section, the heterogenous vaccination strategy was applied. To eliminate your concerns, and according to the suggestions of Reviewers #2 and #3, we have added an analysis of adverse effects of the COVID-19 vaccine after each injection. Type of vaccine was included in the statistic model.

Second, Authors must report how many individuals were initially vaccinated, in order to understand whether the 1322 may acknowledged as representative or not of the original population. 

Answer: Thank you for your suggestion, the following sentence was added to Results section

“During the study period, 1467 individuals were vaccinated with booster dose and invited to participate in the study. A total of 1322 persons (90.1%) agreed to participate and were eligible for selection.”

The minor shortcomings are mainly related to the introduction, as (page 1, since "A primary ..." to "...ever") it mainly deal with the background of late 2021, but an up-ta-date description of the 2022 vaccination campaign must be provided and accordingly discussed.

Answer: Thank you for your suggestion. It was reworded and the statistic was also updated as following:

“A primary vaccination course consists of 2 doses to achieve protective antibody titers. However, recent reports increasingly show that the effectiveness of the COVID19 vaccine is reduced within 6 months after the deployment of the primer dose [3-5]. The COVID-19 vaccine booster dose could further restore and enhance protection that may have decreased over time followed the primary series immunization. Individuals with their updated COVID-19 vaccine status are protected best from severe COVID-19 disease. Therefore, booster shots are essential in the context of the ongoing pandemic due to new variants of concerns. As of August 04, 2022, 12.4 billion doses of the vaccine have been administered worldwide, and 6.3 million doses are given each day. Over 67.2% of the world's population has received at least one dose of the COVID-19 vaccine. It can be said that this has been the largest vaccination campaign on a global scale ever.”

Round 2

Reviewer 1 Report

To the authors: I noticed the changes made in tables 1 and 2. These add value to the manuscript, but I wanted an answer to the question: Was there any difference in terms of the type, frequency, or intensity of adverse effects between subjects vaccinated with different types of vaccines? (AstraZeneca, Sinopharm, Moderna, Pfizer)?

This study is among the few that I have seen that includes subjects vaccinated with different types of vaccines. Thus there would be an opportunity to compare these groups more deeply both in terms of adverse effects (in this study) and the duration of the immune response (in further studies).

Author Response

To the authors: I noticed the changes made in tables 1 and 2. These add value to the manuscript, but I wanted an answer to the question: Was there any difference in terms of the type, frequency, or intensity of adverse effects between subjects vaccinated with different types of vaccines? (AstraZeneca, Sinopharm, Moderna, Pfizer)?

This study is among the few that I have seen that includes subjects vaccinated with different types of vaccines. Thus there would be an opportunity to compare these groups more deeply both in terms of adverse effects (in this study) and the duration of the immune response (in further studies).

Answer: Thank you very much for your comments. We corrected our analysis according to your suggestion. Please see table 4 and Supplementary tables 1, 2 and 3.

Author Response

Thanks for the correction! I want to comment on some things that I overlooked at first. However, please remember that I am not from an English-speaking country regarding the English wording of the proposal, so it is only for reference.

Answer: Thank you very much for your comments. The manuscript was revised according to your suggestions.

Title.

  1. It would be better to include the location and time of the project. For example, "In Vietnam" or "Just after the start of Booster administration.”

Example)

Short-term adverse effects immediately after the start of COVID-19 booster vaccination in Vietnam.

Answer: It was corrected.

Abstract

  1. I see that Tables 2 and 4 have been added. Are these not mentioned?

Answer: We added it in the abstract

Introduction. The background and objectives of this study have been clarified. I understand it better than before.

Answer: Thank you

  1. Line 78

Shouldn't "et al," be "et al.,”

Answer: It was corrected

  1. Line.85.

There seems to be a period missing after reference 14.

Answer: It was corrected

Results and Tables

  1. Lines 163-174.

If it is a supplemental table, I think it would be better to use limitations instead of Results.

For example, how about the following contents?

Supplementary Tables 1-3 show that for the first and second doses, the occurrence of adverse reactions decreased significantly with each additional year of age. It is unclear whether the adverse reactions ceased with increasing age or whether the subjects forgot about the adverse reactions that occurred during the first and second doses. It is also possible from Supplementary Tables 1-3 and Table 4 that as age increases, subjects may forget past adverse reactions and remember the most recent adverse reactions. Thus, we cannot significantly rule out the possibility of a recall bias as age increases. In the future, we would like to consider a system to record adverse reactions for each vaccine, etc.

Answer: Thank you very much for your comments, according to the Reviewers’ comments (Round 1), this analysis was added, and we think it is the result of the study. However, since that was not the main objective of our study, we have included it in the supplementary data.

Your suggestion is very interesting, We added it to the Limitation section to discuss the results we observed.

  1. Supplementary Table and Table

I would like to see annotations outside the columns. For example, the age was used as is. Also, what was used for multivariate analysis?

Answer: It was corrected in Tables and footnotes

  1. Lines 166-168.

I am stuck on the expression "inversely proportional." It would mean that "for the first dose, the incidence of adverse reactions decreased significantly with each additional year of age.”

Answer: It was corrected

  1. Supplemental table1-3 and Table 4

Are there any missing parts?

Answer: No, only variables with p-value <0.2 were introduced in the model. We added it to footnote of each table

  1. For P-values, please unify the number of decimal places.

Answer: The smaller the p-value presents the higher significance of the statistical analysis. Please allow us to keep it as it is.

Reviewer 4 Report

Authors have properly and extensively addressed all my concerns; therefore, I've no further requests.

Author Response

Authors have properly and extensively addressed all my concerns; therefore, I've no further requests.

Answer: Thank you very much

Round 3

Reviewer 1 Report

I have noticed that the authors included data regarding differences in frequency of adverse effects between subjects vaccinated with different types of vaccines in table 4 and Supplementary tables 1, 2, and 3.

In conclusion, after reading the revised manuscript, I find that the authors have fulfilled all the requested suggestions and significantly improved the paper. As a consequence, I am glad to recommend the article for publishing in the Vaccines.  

Author Response

I have noticed that the authors included data regarding differences in frequency of adverse effects between subjects vaccinated with different types of vaccines in table 4 and Supplementary tables 1, 2, and 3.

In conclusion, after reading the revised manuscript, I find that the authors have fulfilled all the requested suggestions and significantly improved the paper. As a consequence, I am glad to recommend the article for publishing in the Vaccines.  Answer: Thank you very much